# Influence of Calcination Temperature and Amount of Low-Grade Clay Replacement on Mitigation of the Alkali–Silica Reaction

**DOI:** 10.3390/ma16083210

**Published:** 2023-04-19

**Authors:** Daria Jóźwiak-Niedźwiedzka, Roman Jaskulski, Kinga Dziedzic, Aneta Antolik, Mariusz Dąbrowski

**Affiliations:** 1Institute of Fundamental Technological Research, Polish Academy of Sciences, Pawińskiego 5B, 02-106 Warsaw, Poland; kdzie@ippt.pan.pl (K.D.); aantolik@ippt.pan.pl (A.A.); mdabrow@ippt.pan.pl (M.D.); 2Department of Civil Engineering, Wrocław University of Environmental and Life Sciences, Grunwaldzki Sq. 24, 50-363 Wrocław, Poland; roman.jaskulski@upwr.edu.pl

**Keywords:** alkali–silica reaction (ASR), expansion, low grade calcined clay, mitigation, Fratini test, microscopic analysis

## Abstract

Results of experimental investigation on the mitigation of alkali–silica reaction (ASR) by low-grade calcined clay are presented. Domestic clay with an Al_2_O_3_ content equal to 26% and SiO_2_—58% was used. The calcination temperatures were as follows: 650 °C, 750 °C, 850 °C and 950 °C, which were chosen much more widely than presented in previous studies. Pozzolanity of the raw and calcined clay was determined with the Fratini test. The performance of calcined clay to mitigate ASR was evaluated according to ASTM C1567 using reactive aggregates. A control mortar mixture was prepared with 100% Portland cement (Na_2_O_eq_ = 1.12%) as a binder with reactive aggregate, and test mixtures were made with 10% and 20% of calcined clay as a cement replacement. The microstructure of the specimens was observed on the polished sections using scanning electron microscope (SEM) operated in backscattered mode (BSE). The results of expansion of mortar bars with reactive aggregate showed that replacing cement with calcined clay reduced the expansion of the mortar bars. The greater the cement replacement, the better results in terms of ASR mitigation. However, the influence of the calcination temperature was not as clear. The opposite trend was found with the use of 10% or 20% calcined clay.

## 1. Introduction

Alkali–silica reaction (ASR) is a complex chemical reaction between alkalis present in concrete and certain types of reactive silica minerals in aggregates, which can lead to significant expansion and cracking of concrete over time. This phenomenon aroused the attention of researchers, in order to develop the methodologies for its mitigation and, most importantly, possible prevention. So far, despite the efforts made and actual recommendations to avoid the occurrence of this phenomenon in new concrete constructions [1,2,3], research on the efficient ways to control its expansive consequences is still needed [1,4,5]. Previous research on the use of fly ash (FA), ground granulated blast furnace slag (GGBS) or silica fume (SF) as supplementary cementitious materials (SCM) to mitigate alkali–silica reaction was extensively studied, and they are still being debated and remain an interesting research topic [1,6,7,8,9]. However, the supply of the traditional SCMs, such as GGBS—a by-product of pig iron production in blast furnaces and FA—generated by burning coal to produce electricity will gradually decrease due to changes in the technological process of steelmaking and the shift away from coal as a fuel in power plants. Therefore, the possibility of using other types of SCMs is increasingly investigated, which include calcined clays.

Calcined clays are manufactured by heating raw material under high temperatures ranging from 600 °C to 800 °C [10]; however, typically, raw clay is calcined between 700 °C and 800 °C [11]. Kaolin-containing clays are the primary raw materials used for manufacturing calcined clay. The best known and best studied material belonging to this group is metakaolin, which is a product of kaolinite calcination [10,11,12]. Kaolinite is dehydroxylated in the temperature range 400–600 °C, which leads to a material with a more disordered structure known as metakaolin [12,13]. One of the main benefits of using metakaolin in concrete is increasing its durability due to the substantial pore refinement, resulting in a modification of the water transport properties and the diffusion rate of harmful ions [14]. Additionally, the role of metakaolin in reducing the expansion due to alkali–silica reaction was investigated [11,12,13,14,15,16,17,18]. Previous research [16,17] showed that metakaolin reduces the concentrations of hydroxide ions (OH^−^), potassium ions (K^+^) and sodium ions (Na^+^) in the pore solution of concrete, which are the key factors that contribute to the development of ASR. The high aluminum content in metakaolin was also shown to contribute to ASR reduction. By reducing these ions, metakaolin helps to prevent the formation of the expansive gel and, thus, mitigates the risk of ASR. Aquino et al. [17] showed that the ASR products have lower levels of calcium when metakaolin was present. A study by Li et al. [16] indicated that chemical composition and mineral phase of calcined clays had a significant influence on the efficacy of ASR mitigation. The effectiveness of calcined clays varied depending on their source and quality. However, the efficacy of calcined clays from different sources and qualities for ASR mitigation is still limited. Moreover, although ASR gel was found in the analyzed mixtures, no significant differences in the gel composition were found between them [16], whereas the conducted preliminary studies showed significant differences in the composition and morphology of ASR gel depending on the use of variable amounts of low-kaolinite calcined clay [19]. The Al_2_O_3_ content in clay calcinated in temperature 850 °C was 21%, well below the previously tested calcined clays (40–45%) [16]. However, due to the limited availability of kaolin and the high demand for it in other industries, clays with a low content of kaolinite or other mineral composition were also tested for their possible use in cement and concrete technology [20,21,22,23]. Calcination temperature is an important factor that can influence the effectiveness of low-grade clay as an SCM in mitigating ASR. At lower temperatures, the pozzolanic properties of the clay may not be fully revealed, resulting in lower reactivity and, therefore, less effective ASR mitigation. On the other hand, at higher temperatures, the clay may become overburnt and lose its pozzolanic activity due to partial sintering, leading to reduced ASR mitigation. Preliminary research conducted by Fernandez et al. [23] suggested that the activation temperature did not have a significant impact on the pozzolanic properties of most calcined clay minerals, except for calcined montmorillonite. The study found that increasing the calcination temperature had a strong effect on the particle size distribution and specific surface area of calcined montmorillonite. This is because montmorillonite is a layered clay mineral that has a higher degree of structural order compared to other clay minerals. As a result, it requires a higher activation energy to undergo the necessary structural changes during calcination, which can result in changes to its particle size and specific surface area. Linhares Jr. et al. [24] stated that the calcination temperatures should be adjusted according to the degree of purity and crystallization of the clay, and they generally alternate between 500 and 800 °C.

The studies presented in this paper focus on the effect of the calcination temperature of locally available low-grade calcined clay in terms of its ability to control the expansion by the alkali–silica reaction. The scope of experimental work includes the characteristics of raw clay and clay calcined at four temperatures: 650, 750, 850 and 950 °C, assessment of the effectiveness of calcined clay in ASR mitigation according to ASTM C1567 [25] and microstructural analysis. The pozzolanic activity of both raw and calcined clay was also investigated by performing a test based on Fratini’s studies [26,27,28].

## 2. Materials

Raw clay was used with 26% Al_2_O_3_ content as measured by X-ray fluorescence (XRF, Table 1). Loss on ignition (LOI) was equal to 9.20%. The main associated minerals, except for kaolinite, muscovite and illite, found in the clay by XRD was quartz (see Figure 1). The differences in the XRD diffractograms were due to the dehydroxylation of kaolinite, muscovite and illite (weight loss up to 700 °C). Portland cement CEM I 42.5R was used for mortar manufacturing. Na_2_O equivalent was equal 1.12% and Blaine’s specific surface 546 m^2^/kg. The MgO soundness evaluated through Le Chatelier test was 0 mm. Nine mortar compositions were tested: reference without any additions and 10% and 20% of cement replacement by clay calcined in four different temperatures (650 °C, 750 °C, 850 °C and 950 °C).

For each composition, three 25 mm × 25 mm × 285 mm mortar bars were prepared using reactive aggregate (expansion 0.18% after 14 days according ASTM C1260 [29]).

## 3. Methods

### 3.1. Materials Characterization

Clay calcination was carried out in a laboratory chamber furnace with a heating capacity of 8.8 kW. Prior to loading into the furnace, the clay was dried to constant weight at 110 °C for 2 days, and then, was ground and sieved through a 0.125 mm mesh. The clay was calcined in alumina crucibles, each containing approximately 350 g of loosely filled material. The crucibles were filled to two-thirds to avoid clay spilling out due to autobarbotage (Barbotage (*colloq.* ‘bubbling’) is the passing of gas bubbles through a liquid in order to increase the contact area or the surface area available for reaction. The word ‘autobarbotage’ was used to describe a similar process occurring in a finely ground suspension of a solid phase in a gas, where the source of the gas is a reaction taking place in the solid phase.) during the calcination process. The heating ramp was, in all cases, 4 °C/min until reaching a temperature of about 150 °C and 12 °C/min after this threshold was exceeded (this procedure was a result of the furnace software to prevent excessive burnout of the heater). After the target calcination temperature was reached, it was maintained for 60 min, after which the material was allowed to cool in the oven to a temperature below 50 °C. After cooling, the calcined clay was stored in sealed plastic bags until testing.

X-ray diffraction (XRD) was used to estimate the main minerals in the investigated raw and calcined clays. In the described method, the specimens were first sieved through a 0.045 mm sieve to ensure a uniform particle size distribution. XRD diffractograms were obtained at room temperature with a diffractometer (Bruker, Karlsruhe, Germany) (Cu Kα radiation source) equipped with a Göbel mirror and a GADDS 2D detector system. The operation parameters of the equipment, such as the voltage and current, were set to 40 kV and 40 mA, respectively. The diffraction patterns were collected over a 2θ range from 5° to 70° with a 1°/min step, using a flat plane geometry. The obtained data were then analyzed using the Diffrac.Eva V5.2 software to identify the minerals present in the specimens.

The specific surface area (SSA) of the calcined clays was determined by gas adsorption using the Brunauer–Emmett–Teller (BET) method, with a Quantachrome Autosorb IQ analyser (Graz, Austria). Nitrogen (N_2_) was used as analysis adsorptive gas. The procedure involved introducing approximately 0.5 g of the powder material into analysis tubes and degassing them for 16 h at 105 °C under vacuum to remove any moisture or other volatile impurities. After degassing, the tubes were immersed in a liquid nitrogen bath at 77.3 K, and gas adsorption measurements were performed. Specifically, 25 points of adsorption and 20 points of desorption were measured at different relative pressures (P/P0) ranging from 0.05 to 1. These measurements were used to calculate the specific surface area of the material using the BET equation.

The pozzolanic activity of both raw and calcined clays was determined by performing the Fratini test [26,27,28,30]. To a HDPE container with 100 mL of distilled water preheated to 40 °C, a portion of the material mixture was added. The portion consisted of 14 g of CEM I 42,5 R Portland cement and 6 g of the tested clay. The contents were then mixed and the container was put into a thermostat at temperature of 40 °C ± 2 °C for 15 days. After this time, 50 mL of the filtrated solution was collected and two titrations were performed: first, with hydrochloric acid in the presence of methyl orange to determine the concentration of OH^−^ ions, and second, with tetrasodium edetate in presence of ammonium purpurate in order to determine the concentration of Ca^+^ ions. The results, after calculation, were transferred to a diagram in order to determine whether the tested sample exhibited pozzolanic activity. In addition, the so-called Fratini pozzolanity was calculated, which was the difference between the tested concentration of Ca^+^ ions and the maximum concentration of these ions at a given concentration of OH^−^ ions. Three replicates were performed in each case, and the results were averaged, rejecting outliers if necessary.

### 3.2. Accelerated Mortar Bar Tests

The accelerated mortar bar test was performed according to ASTM C1567 [25] procedure. The aggregate was processed by crushing and sieving to the appropriate gradation. The mortars were mixed and molded within a total elapsed time of less than 2 min. The mortar bars were kept at 23 ± 1 °C and RH ≥ 95%, for 24 h. After 24 h, the specimens were demolded, submerged in distilled water and heated to 80 °C. After 24 h, the bars were removed from water, measured and immersed in 1M NaOH solution previously heated to 80 °C. The specimens were stored in thermostatic chamber at 80 °C, subsequent expansion readings were measured at least three times during the test.

### 3.3. Microstructural Investigation Using SEM

The specimens for microstructural investigation were obtained by slicing post-mortem specimens with a slow speed diamond saw. The specimens 50 × 30 × 15 mm were dried at 50 °C for 3 days, and they were vacuum-impregnated with a low-viscosity epoxy. The thin sections were prepared by grinding and polishing the aggregates up to desired thickness. For polishing silicon carbide papers, up to 1200 mesh were used. Final thickness was obtained by polishing with diamond paste 6-3-1-0.25 μm on cloth, to obtain a mirror finish [31]. The specimens were prepared for examination using a scanning electron microscope (SEM) equipped with an energy-dispersive X-ray spectroscopy (EDX) analyzer. Specifically, each specimen was prepared so that the polished face to be examined was a cut surface from the middle of the bar. The specimens were then coated with carbon to make them conductive, and a strip of conductive tape was attached to each specimen. The SEM-EDX analysis was performed using a JEOL JSM-6380 LA SEM in the backscatter mode with an acceleration voltage of 15 kV. Genesis Spectrum 6.2 by EDAX Inc. (Tokyo, Japan) as EDS analyzer was used.

## 4. Results and Discussion

The chemical composition of raw clay is shown in Table 1. The relatively high content of K_2_O (1.86%) is noteworthy, proving mainly the presence of illite and/or muscovite, which were confirmed using XRD method (Figure 1). On the other hand, the content of MgO and CaO (0.27% and 0.22%, respectively) which proves the presence of smectites at most was very low. The potential presence of minerals from this group in the investigated raw clay was not confirmed by the XRD test method.

When analyzing XRD patterns (Figure 1) of raw and calcined clays at selected temperatures, three processes are worth noting. The first is the practically complete disappearance of the peaks characterizing kaolinite, which occurs at the lowest calcination temperature, i.e., 650 °C. This temperature is usually considered to be over the upper limit of the process of thermal decomposition of the kaolinite structure, although some sources claim that this decomposition continues even above 750 °C [11]. The second process, which is well visualized on the obtained XRD patterns, is the gradual decomposition of muscovite and illite. The characteristic peaks become less and less visible as the calcination temperature increases. The third process, whose effects are visible on XRD patterns of clay calcined at the highest temperature (950 °C), is the sintering of the material associated with the formation of new crystalline phases. One peak around the 2Theta angle of about 16° was particularly prominent. This is probably a signal of the presence of analcime (15.88°), which rarely forms regular crystals. The XRD patterns showed that the calcination of clay at 650 °C completely dehydroxylated kaolinite, as seen by the persistence of the main kaolinite reflection at 12.36° 2Theta angle. The main muscovite peak at 8.87° 2Theta angle slightly decreased with increasing temperature and at 850 °C, dehydroxylation occurred [21], which is confirmed in the literature [32].

The relationship between specific surface area (SSA) results of calcined clays, measured using nitrogen adsorption, and their calcination temperature is presented in Figure 2. The above results agree with previous studies [14,20], where the SSA decreased with increasing calcination temperature. It is clearly visible that only above 750 °C, the specific BET surface area significantly decreases. Calcination temperatures between 750 and 850 °C significantly decreased the SSA, almost 30%. Such a reduction in, and structural reorganization of, aluminosilicates [33] can limit the pozzolanic reactivity of calcined clay. A study performed by Fernandez et al. [23] showed that the BET specific surface of kaolinite decreased only about 2% (from 24.6978 m^2^/g to 24.1283 m^2^/g) as the calcination temperature increased from 600 °C to 800 °C, whereas the BET-specific surface of montmorillonite decreased about 55% (from 21.386 m^2^/g to 9.7221 m^2^/g) with the same temperature increase. In the conducted test, the BET specific surface decreased about 58% (from 22.5 m^2^/g to 9.4 m^2^/g) as the temperature increased from 650 °C to 950 °C.

The results of the Fratini test are shown in Figure 3 in a diagram with the lime solubility curve drawn in. For all the clay samples tested, the results were below the curve, meaning that they all showed pozzolanic activity. There was also a clear positive effect of calcination on the level of this activity, although it is not clear from the graph shown whether the temperature at which this process was carried out affects the pozzolanic activity obtained. What is clear is the effect of temperature on the pH of the solution, which is determined by the position of the results relative to the *X* axis. The alkalinity of the solution with raw clay and clay calcined at 650 °C was similar, after which it reached its minimum at a calcination temperature of 750 °C. It then increased markedly with increasing temperature. This increase in alkalinity may explain the release of an additional portion of alkali from the temperature-defected clay structure.

To determine the effect of temperature on the pozzolanic activity of the calcined clay more precisely, the value of the so-called Fratini pozzolanity was calculated. This value indicates how much the calcium ion concentration decreased in relation to the maximum value it can reach at a given pH level. For example, in the situation indicated by the green arrow in Figure 3, the CaO concentration determined from the lime solubility curve was 11.49 mmol/L, while the concentration determined by the Fratini test was 6.09 mmol/L. After subtracting these two values from each other, the resulting Fratini pozzolanity value was 5.40 mmol/L. The results of these calculations are presented in Figure 4, which shows that the maximum pozzolanic activity was reached by the calcined clay at 750 °C. Subsequently, at 850 °C, the activity decreased, although it still remained above the level obtained at 650 °C. A further increase in temperature had little effect on the Fratini pozzolanity value.

Results of the mortar-bars expansion measurement as function of their storage time in 1N NaOH solution at 80 °C are shown in Figure 5. The dotted red line indicates maximum admissible value of linear elongation for specimens with aggregates that are not reactive.

The effect of low-grade calcined clay on the expansion of mortars is visible. The replacement of cement by clay lowered the elongation of the mortar bars. The replacement of 10% of cement resulted in a reduction in expansion by 19 ± 7% and the replacement of 20%–68 ± 5%. It should be noted that the mortar bars containing 20% of calcined clay, despite showing significantly lower expansion levels than the reference specimens and the specimens with 10% of calcined clay, did not show a marked flattening of the expansion curve at the end of the test. This is probably due to the fact that the analyzed mortar bars were immersed in 1 N NaOH, which can provide a constant supply of alkali. In the study, specimens were tested according to ASTM C1567 [25] to compare the influence of various calcination temperatures on the ASR mitigation and not to draw conclusions about absolute effectiveness of low-grade calcined clay.

In mortar bars with a replacement of 10% of cement, the lowest expansion was obtained in specimens containing low-grade clay calcined at a temperature of 650 °C. The influence of the clays calcined in the other temperatures was very similar, the expansion was 0.16–0.17% compared to 0.14% for specimens with clay calcined at 650 °C. In contrast, the opposite trend was noticeable when 20% of calcined clay was used. The lowest expansion was achieved by mortar bars containing clay calcined at 950 °C (0.05%), while the highest was observed in those containing clay calcined at 650 °C (0.07%). In both cases, the difference between the extreme values of the expansion of the mortar bars, i.e., for the replacement of 10% and 20% of cement with low-grade calcined clay, was approximately 20% and 40%, respectively. This finding may be a direct result of the effect of the calcination temperature on the specific surface area of the obtained clay particles (Figure 2 and Figure 6). A similar observation was found by Khan et al. [34], Ramjan et al. [35] and Aydın et al. [36]. Khan et al. [34] revealed that at low percentages (10%), ASR expansion in both very fine ash and ultra-fine ash mortars was higher compared to the corresponding fly ash mortar and the opposite behavior was observed at high percentages (20% and 30%). Ramjan et al. [35] found that the mortars containing bagasse ash with higher fineness exhibited lower expansion due to ASR than the mortars with this kind of ash with lower fineness. Aydın et al. [36] confirmed that partial replacement of cement with fly ash reduced the ASR expansions of mortar bars, but this positive effect was found for more than 10% cement replacement. However, Harish and Rangaraju [37] found that in the case of high-lime fly ash, the effect of fineness could not be clearly resolved and Oruji et al. [38] found that a portlandite consumption depended on pulverized bottom ash fineness levels.

After the accelerated expansion test specimens, mortar bars were prepared for ASR gel analysis by SEM-EDS. The evidence of alkali–silica reaction—cracking and microcracking in aggregate and in cement matrix—occurred in the reference specimens as well as in the specimens with 10% and 20% of calcined clay (see Figure 7, Figure 8 and Figure 9). The chemical composition of ASR gel filling the air-voids is presented in Figure 7. The photos show how the degradation of aggregate grains decreases with increasing the addition of calcined clay. The ASR is not eliminated but the damage is visibly reduced. The differences are also visible in the composition of the ASR gel. The Si peak is of similar height for all specimens; however, the Na and K peaks decrease with the appearance of the Al and Mg peaks.

The influence of the calcination temperature on the ASR mitigation is a complex phenomenon which concerns both differences in the mineral composition and in the clay specific surface area. It is known that calcined clays with the most amorphous structures showed better efficacy in controlling expansion caused by ASR [16]. A high quality calcined clay—metakaolin—contains a high content of alumina and silica (about 40–45% and 52–55%, respectively [13]), which reacts with Ca(OH)_2_ in cement to form C-A-S-H. The analyzed low-grade clay contained similar content of SiO_2_ (58%) but much lower content of Al_2_O_3_ (26%, Table 1), which may influence the C-A-S-H formation. Similar to the fineness levels of fly ash [38] and metakaolin [12,22], the influence of the SSA in calcined clays may play an important role in lowering Ca/Si ratio in CSH and in portlandite consumption. The next study will focus on the detailed microscopic characteristics of the ASR gel and the influence of the Al content on its mechanical parameters.

Additionally, worthy of further investigation is the phenomenon of “inverse effectiveness” in ASR expansion mitigation of clay calcined at different temperatures, depending on the replacement rate of cement with the clay. The clay calcined at 950 °C showed the lowest effectiveness when 10% of the cement was replaced with it, while at an exchange rate of 20%, it proved to be the most effective. This is surprising, as this clay probably releases the largest portion of additional alkali from its defected structure, which should work against it at higher exchange rates. It is worth exploring the hypothesis that the level of expansion due to ASR may be influenced in some way by the amount of free calcium hydroxide available in the material.

## 5. Conclusions

Based on the research carried out, the following conclusions can be drawn:Dehydroxylation of low-grade clay was dependent on the calcination temperature. The favorable temperature for kaolinite dehydroxylated was 650 °C and for muscovite/Illite was 850 °C.The relationship between specific surface area of calcined clay, measured using nitrogen adsorption and its calcination temperature was found.There is a clear effect of clay calcination temperature on its pozzolanic activity as measured by the Fratini pozzolanity value, and the highest pozzolanic activity was shown by clay calcined at 750 °C. However, there is no simple and straightforward translation of the pozzolanic activity measured by the Fratini test into the ability to mitigate the ASR expansion.The study found that the use of low-grade calcined clay can have a beneficial effect on reducing alkali–silica expansion in mortars, regardless of the percentage of substitution with cement. However, the results also indicate that at the lower percentage of substitution (10%), the expansion results did not meet the limits specified in ASTM standards.The ASR expansion results showed the opposite effect when 10% or 20% low-grade calcined clay was used, which may be due to different Ca/Si ratios and portlandite consumption depending on the specific surface area of the calcined clay.

## Figures and Tables

**Figure 1 materials-16-03210-f001:**
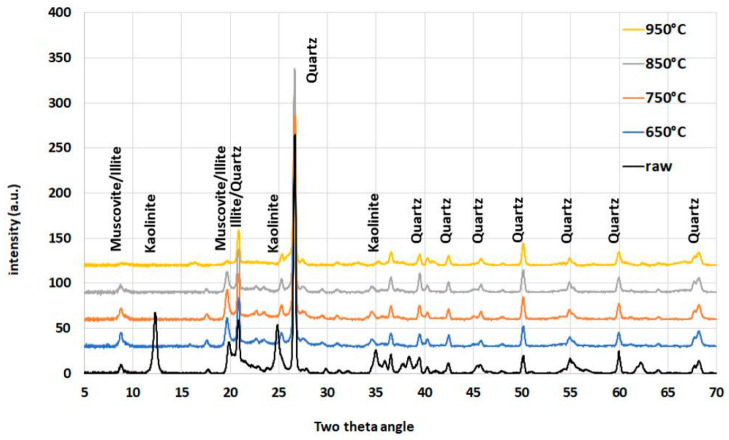
Mineral composition of the clay depending on the calcination temperature.

**Figure 2 materials-16-03210-f002:**
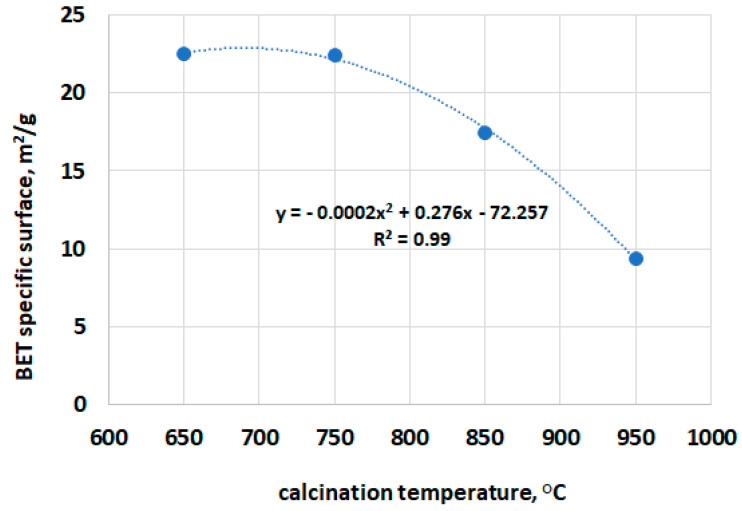
Relationship between SSA of the calcined clay determined by BET method vs. calcination temperature.

**Figure 3 materials-16-03210-f003:**
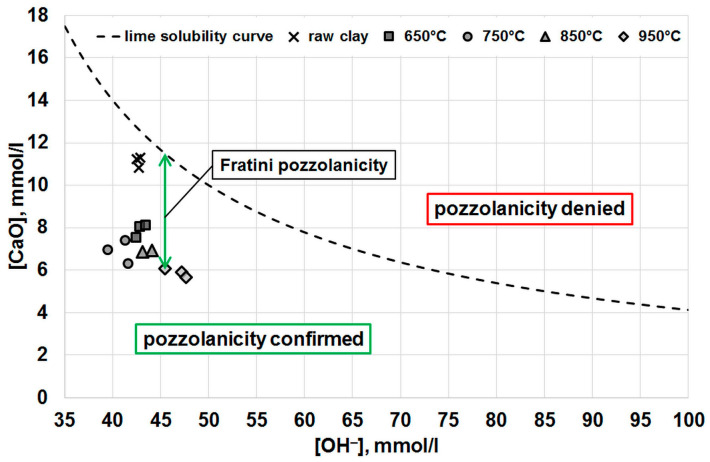
Results of the Fratini pozzolanity test.

**Figure 4 materials-16-03210-f004:**
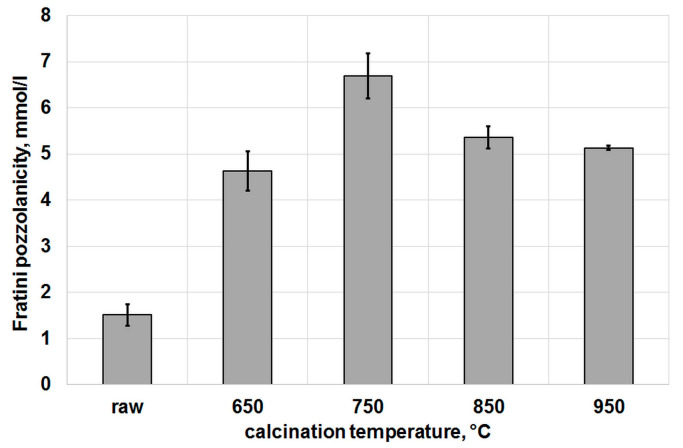
Values of the Fratini pozzolanity vs. calcination temperature.

**Figure 5 materials-16-03210-f005:**
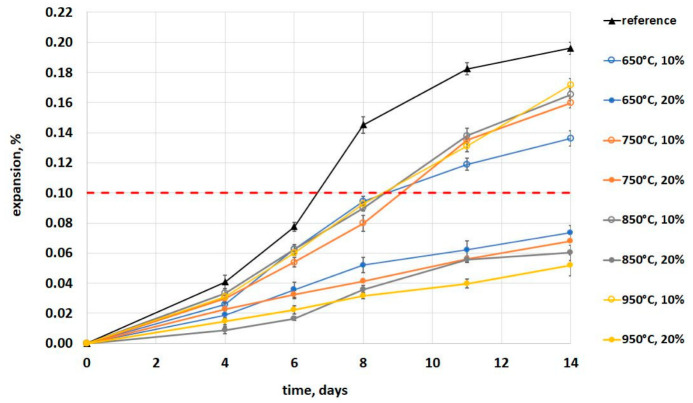
Results of the mortar bars expansion stored in 1N NaOH and at 80 °C.

**Figure 6 materials-16-03210-f006:**
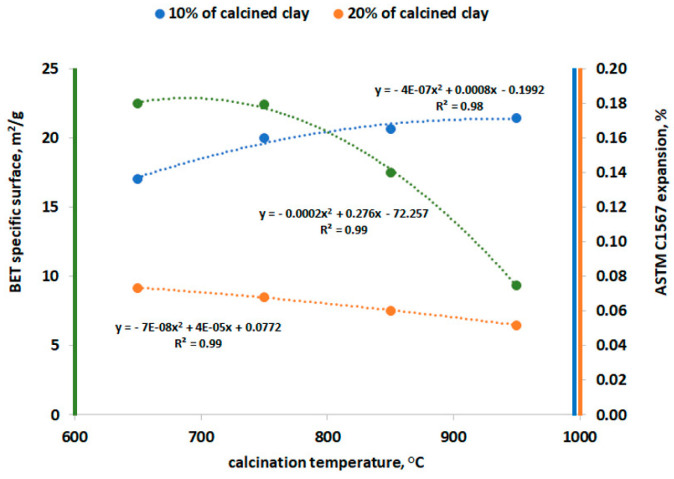
Results of the SSA of the calcined clay determined by BET method and expansion according to ASTM C1567 depending on calcination temperature.

**Figure 7 materials-16-03210-f007:**

Cracking and microcracking in aggregate and in cement matrix in analyzed specimens: (**a**) reference, 0% of calcined clay, (**b**) 10% of clay calcined at 850 °C, (**c**) 20% of clay calcined at 850 °C.

**Figure 8 materials-16-03210-f008:**
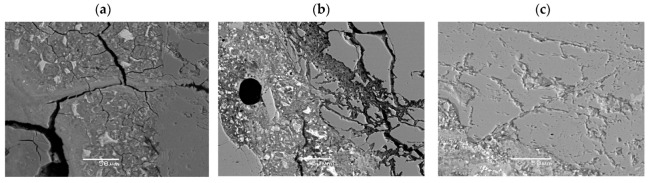
Cracked aggregate and ASR-gel formed in analyzed specimens: (**a**) reference, 0% of calcined clay, (**b**) 10% of clay calcined at 850 °C, (**c**) 20% of clay calcined at 850 °C.

**Figure 9 materials-16-03210-f009:**
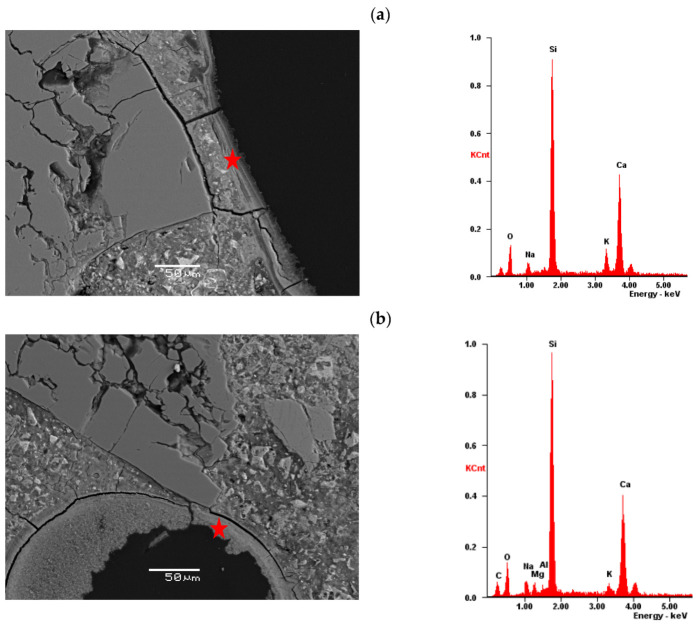
ASR-gel formed in air-voids in analyzed specimens: (**a**) reference, 0% of calcined clay, (**b**) 10% of clay calcined at 850 °C, (**c**) 20% of clay calcined at 850 °C; a star represents the location of the EDS analysis.

**Table 1 materials-16-03210-t001:** Chemical composition of raw clay determined by XRF method (wt %).

Constituent	SiO_2_	TiO_2_	Al_2_O_3_	Fe_2_O_3_	MnO	MgO	CaO	Na_2_O	K_2_O	P_2_O_5_	SO_3_	Cl	F
Content, %	58.23	2.04	26.18	1.53	0.03	0.27	0.22	0.13	1.86	0.06	<0.01	0.01	0.03

## Data Availability

The article is an extension of the research presented at the 10th MATBUD’2023 Scientific-Technical Conference “Building Materials Engineering and Innovative Sustainable Materials”, Cracow, Poland, 19–21 April 2023, https://www.mdpi.com/2673-4605/13/1/15 accessed on 11 April 2023.

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
