# Peer review of "Influence of Calcination Temperature and Amount of Low-Grade Clay Replacement on Mitigation of the Alkali–Silica Reaction"

_materials, 2023, doi:10.3390/ma16083210_

Round 1
Reviewer 1 Report
I enjoy reading the manuscript but there are several points that need to be improved. My comments and corrections were made in an annotated version of the submitted PDF file
The samples preparation and methods used need to be better described, especially for the SEM-EDS analysis.
I find it surprising that the raw clay has pozzolanic activity. Do you have any explanation for this result?
An example of the calculation of Fratini pozzolanity value should be presented.
The influence of SSA of calcined clays in ASR mitigation should also be better explained.
Some conclusions are not supported by the presented results (e.g. the availability of Al2O3 in the amorphous phase!).

Author Response
Dear Reviewer 1,
We highly appreciate your comments concerning our manuscript. We have carefully studied your comments and have introduced appropriate corrections, which we hope will be met with your approval. Thank you for your time spent in reviewing our manuscript. At the same time, we would like to point out that the text of the manuscript has been modified as suggested by Reviewer 2 and Reviewer 3.
Please, find below our detailed answers to your comments and suggestions.
Sincerely,
Daria Jóźwiak-Niedźwiedzka
Reviewer 1: Purple text has been used to show the corrections made in the manuscript.

Reviewer 2 Report
The authors studies the effect of clacination temperature and clays additions on the extent of the alkali-silica reaction. This work is well organized and presented and the conclusions are sound. I think this work is publishable in materials after considering the following points.
1. Line 92, please explain what is the loss of ignition means?
2. Line 110, for how long the clay was dried at 110 oC?
3. Line 175-180, the authors mentioned that kaolinite disappeared on heating the sample to 650 oC based on the peak at about 2 theta of 17 degrees, however, kaolinite peak at 35 degrees persist up to 950 oC. Please make this clear.
4. Line 185-187, can the authors provide a reference for XRD pattern of the emerged phases (i.e., analcime and plagioclase).
Author Response
Dear Reviewer 2,
We highly appreciate your comments concerning our manuscript. We have carefully studied your comments and have introduced appropriate corrections, which we hope will be met with your approval. Thank you for your time spent in reviewing our manuscript. At the same time, we would like to point out that the text of the manuscript has been modified as suggested by Reviewer 1 and Reviewer 3.
Please, find below our detailed answers to your comments and suggestions.
Sincerely,
Daria Jóźwiak-Niedźwiedzka
Reviewer 2: Green text has been used to show the corrections made in the manuscript.

Reviewer 3 Report
The presented work is both significant and reliable. However, the following suggestions are given for potential enhancements:
1. How does the calcination temperature of locally available low-grade calcined clay affect its ability to mitigate alkali-silica reaction?
2. What is the chemical composition and mineral phase of calcined clays that have a significant influence on the efficacy of ASR mitigation?
3. How do the pozzolanic properties of low-grade clay change with calcination temperature, and how does this affect its effectiveness in mitigating ASR?
4. At what temperature did the complete disappearance of the peaks characterizing kaolinite occur during the calcination process, and what are the implications of this observation?
5. How does the specific BET surface area of the calcined clay change as the calcination temperature increases, and how do these results compare to previous studies?
6. Could you discuss the implications of the findings for future research or practice? What are the assumptions made in the statistical analysis of the study data?
7. The following references may be helpful:
a. A mix-design procedure for alkali-activated concrete based on the concept of reactive modulus. T Xie, X Zhao. Handbook of Advances in Alkali-Activated Concrete, 15-40;
b. Prediction of ultimate condition of FRP-confined recycled aggregate concrete using a hybrid boosting model enriched with tabular generative adversarial networks. XY Zhao, et al. Thin-Walled Structures 182, 110318;
c. A review on durability of alkali-activated system from sustainable construction materials to infrastructures. W Li, et al. ES Materials & Manufacturing 4 (2), 2-19.
Author Response
Dear Reviewer 3,
We highly appreciate your comments concerning our manuscript. We have carefully studied your comments and have introduced appropriate corrections, which we hope will be met with your approval. Thank you for your time spent in reviewing our manuscript. At the same time, we would like to point out that the text of the manuscript has been modified as suggested by Reviewer 1and Reviewer 2.
Please, find below our detailed answers to your comments and suggestions.
Sincerely,
Daria Jóźwiak-Niedźwiedzka
Reviewer 3: Blue text has been used to show the corrections made in the manuscript.

Round 2
Reviewer 1 Report
I thank the authors for their answers and clarifications to my questions.
I noticed that they introduced most of the corrections/suggestions I made, but in the case of the captions of figures 7 to 9 I think they could have introduced the text I suggested. I also suggested to the authors that they delete the reference to enlargement in the legend, which they did in figures 7 and 8, but not in figure 9. Please review my suggestions in the annotated version I previously sent.
After that, I think the article will be ready to be accepted for publication.
Author Response
Dear Reviewer 1,
Thank you for your very thorough review of the article.
Corrections have been made.
Kind regards,
Daria Jóźwiak-Niedźwiedzka
Reviewer 2 Report
The authors have addressed my comments and I approve the current version for publication in materials
Author Response
Dear Reviewer 2,
Thank you very much for the detailed review of the article.
Kind regards,
Daria Jóźwiak-Niedźwiedzka